# Assessment of the Performance of Obesity Measures in the Screening of Hypertension in a South African Adolescent Population

**DOI:** 10.3390/children10091520

**Published:** 2023-09-07

**Authors:** Godwill Azeh Engwa, Paul Chungag Anye, Nandu Goswami, Benedicta Ngwenchi Nkeh-Chungag

**Affiliations:** 1Department of Biological and Environmental Sciences, Faculty of Natural Sciences, Walter Sisulu University PBX1, Mthatha 5117, South Africa; nandu.goswami@medunigraz.at (N.G.); bnkehchungag@wsu.ac.za (B.N.N.-C.); 2MBCHB Programme, Faculty of Health Sciences, Walter Sisulu University PBX1, Mthatha 5117, South Africa; 214023362@wsu.ac.za; 3Gravitational Physiology and Medicine Research Unit, Division of Physiology and Pathophysiology, Otto Loewi Research Center of Vascular Biology, Immunity and Inflammation, Medical University of Graz, 8036 Graz, Austria; 4Department of Health Sciences, Alma Mater Europaea, 2000 Maribor, Slovenia; 5College of Medicine, Mohammed Bin Rashid University of Medicine and Health Sciences, Dubai P.O. Box 505055, United Arab Emirates

**Keywords:** obesity, hypertension, adolescents, cut-off values, diagnostic performance

## Abstract

Obesity is known to be one of the most significant risk factors for essential hypertension in childhood. However, whether obesity cut-offs may predict hypertension screening in adolescents remains controversial. This study investigated the performance of obesity cut-off values for the screening of hypertension in a South African adolescent population. In this cross-sectional study, 1144 adolescents aged between 11 and 17 years were recruited from the Eastern Cape Province of South Africa. Anthropometric and blood pressure (BP) parameters including diastolic blood pressure (DBP), systolic blood pressure (SBP), and heart rate (HR) were measured. Assessment of the sensitivity and specific of obesity cut-off values in predicting hypertension was performed using receiver operating characteristic (ROC) analysis. Obesity was positively associated (*p* < 0.01) with hypertension. Obese individuals, based on the predicted obesity using BMI percentile cut-off (pBMI85.2%), were more likely to develop hypertension (odds ratio: 2.070; *p* < 0.001) than their counterparts based on the observed obesity cut-off (pBMI95%) (Odd ratio: 1.748 *p* = 0.004). The area under the curve (AUC) of BMI percentile and WHtR for screening SBP percentile, and DBP percentile and HR as per ROC analysis, was low (<0.65). Equally, the sensitivity and specificity were low (<0.6) for all BP measures (SBP, DBP, and HR). Furthermore, the cut-off values for blood pressure measures, as established by ROC analysis using anthropometric measures, were far below the recommended cut-off values for hypertension screening. The obesity cut-offs for BMI percentile and WHtR established in this populations showed poor performance in diagnosing hypertension even though they were strong predictors of hypertension.

## 1. Introduction

Hypertension continues to be a global problem as the predominant cause of cardiovascular and metabolic-related diseases and a major cause for premature death worldwide [1]. Although hypertension is a commonly observed disease in adults, it is becoming a major concern in young adults and even children. Childhood essential hypertension is a health problem of concern as it can progress to hypertension in adulthood [2,3]. This concern is attributed to the increased hypertension prevalence in children as well as adolescents [4]. According to a systematic review and meta-analysis in 2018, the global pool prevalence of childhood hypertension is about 4% and approximately 9% for prehypertension [5]; meanwhile, in Africa, a systematic review and meta-analysis in 2022 reported a pooled prevalence of 7·45% and 11% for hypertension and elevated blood pressure, respectively, in children [6]. A much higher prevalence has been reported in South Africa, with a systematic review reporting a range from 7.5% to 22.3% for hypertension prevalence in South African children [7]. Other studies in the Eastern cape province of South Africa have shown an even higher prevalence of hypertension in children [8] and adolescents [9].

Although not the sole contributor to the development of hypertension, obesity is known as one of the major predictors of hypertension [10]. Previous studies have reported that obese and overweight children have a higher risk for hypertension compared to non-obese children [11,12]. Various anthropometric indices have been used to screen obesity including measures of general obesity such as mid upper-arm circumference (MUAC), neck circumference (NC), and body mass index (BMI), as well as measures of central obesity such as hip circumference (HC), waist circumference (WC), and their ratios, waist–height ratio (WHtR) and waist–hip ratio (WHR) [13]. There is, however, no consensus on which anthropometric measures are better predictors of childhood hypertension, as certain studies claim measures of central obesity (WHR, WC and WHtR) [14,15] to be superior to general obesity (BMI) [16,17,18] while in other studies, it is not distinct [19]. Moreover, screening hypertension in adolescents and children is more complicated than in the adult population as childhood hypertension is dependent on height, gender, and age-specific references in diastolic blood pressure (DBP) and systolic blood pressure (SBP) [20].

In our previous study [21], we established new cut-off values for obesity screening in adolescents in a South African population. Our findings, based on ROC analysis, showed that BMI percentile (pBMI%) and WHtR were the most suitable measures for obesity screening in adolescents. With the new cut-offs for pBMI% and WtHR, the predicted prevalence of obesity increased compared to the observed prevalence based on the WHO [22] and the CDC [23] cut-off references. However, it remains unclear how these new cut-offs will predict hypertension in this population. Hence, the aim of this study was to assess the relationship between blood pressure and obesity based on these new cut-off values and assess their performance to predict hypertension in adolescents in a South African population.

## 2. Materials and Methods

### 2.1. Study Design and Population

A cross-sectional study which recruited 1144 South African adolescents including 348 males and 796 females aged between 11 and 17 years old was conducted. The participants were recruited from Mthatha, Libode, Alice, and East London of the Eastern Cape Province inhabited by South Africans of African origin.

### 2.2. Ethical Consideration

The study was approved by the Ethics Committee of University of Fort Hare (CH1011SCHU01) on 1 June 2016, and the Faculty of Health Sciences, Walter Sisulu University (Ref No: 112/2018), on 31 October 2018, South Africa. Prior to enrolment into the study, written informed consent was obtained from the parents or legal guardians of the children. Participants’ data was kept confidential in accordance with the South Africa National Data Protection Act.

### 2.3. Blood Pressure Measurements

After resting for 5 min in a quiet room while sitting, participants were fitted with arm-size appropriate cuffs and blood pressure was measured three times at every 2 min interval using an automated sphygmomanometer (HBP-1100, Omron Healthcare Co., Kyoto, Japan). The mean blood pressure (BP) was calculated and then age, sex, and height were used to convert the BP to percentiles according to the guidelines for children (Centre for Disease Control and Prevention-National Health and Nutrition Examination Survey) [24]. Participants were classified as normotensive (NT) when diastolic BP (DBP) and systolic BP (SBP) was <90th percentile; pre-hypertensive (preHT) when DBP and SBP > 90th < 95th percentile; or hypertensive (HT) when DBP and/or SBP ≥ 95th percentile. Mean arterial pressure (MAP) was calculated as per the formula: MAP = (SBP + (2 × DBP))/3.

### 2.4. Anthropometric Measurements

Anthropometric measurements were taken in accordance with the International Standards for Anthropometric Assessments [25]. A wall-mounted Harpenden stadiometer was used to measure height in meters (m). Tanita weight scale (BC1000, Tanita Corporation, Tokyo, Japan) was used to measure the weight (kg), and the height was entered in the device to calculate the body mass index (BMI) expressed as weight/height^2^ (kg/m^2^). BMI, taking into consideration the age, sex and height of the children, was converted to BMI percentiles (pBMI) and classified as underweight: <5th percentile; normal weight: ≥5th to <85th percentile, overweight: ≥85th to <95th percentile; and obese: ≥95th percentile [26]. An anthropometric tape was used to measure the waist circumference (WC), hip circumference (HC), mid-upper arm circumference (MUAC), and neck circumference (NC) in centimetres (cm). WC and height were used to calculate the waist-to-height ratio (WHtR) and obesity was considered above a cut-off value of 0.5, as previously reported [27].

### 2.5. Statistical Analysis

The prevalence of hypertension was calculated as (presence of hypertension/total population) × 100. Predicted obesity was defined as obesity using the new cut-off values established in our previous study [21], while observed obesity is obesity based on the WHO and the CDC reference cut-off values. Statistical Package for Social Sciences (SPSS) software (version 20, IBM SPSS Inc., 2011, Chicago, IL, USA) was used to analyse data. Data were presented as mean ± standard deviation in tables. The mean differences between groups were analysed by independent sample t-test after adjusting for sex and age. Chi-square test was used to assess the association of obesity and hypertension by comparing categorical variables between groups while binary logistic regression was employed to determine the odds ratio of obesity to predict hypertension. Receiver operating characteristic (ROC) curve analysis [28] was used to determine the performance of anthropometric measurements to predict blood pressure and the area under the curve (AUC), an indicator to measure how precise an anthropometric measure distinguishes hypertension. The AUC ranged between 0 and 1. A value of 0.5, indicated by a diagonal line, shows that the anthropometric measure has no predictive performance for hypertension while a value of 1 indicates an ideal or optimal performance. The accuracy of anthropometric measures in predicting hypertension was determined by the specificity and was obtained from the ROC curve. The ROC curve is a plot of the sensitivity (true-positive rate) against 1-specificity (false-positive rate) for each anthropometric measure. The Youden index (value of the largest sum of sensitivity and specificity-1) was used to determine the optimal cut-off value for each anthropometric index (HC, WC, BMI, WHtR, and MUAC). A difference was considered to be significant at *p* ≤ 0.05.

## 3. Results

### 3.1. Characteristics of Study Participants

A total of 193 adolescents out of 1144 were hypertensive giving a prevalence of 16.9%. Among females, 17.1% (136) were hypertensive while 16.4% (57) of males were hypertensive. The population was matched for weight between boys and girls, although boys were slightly taller than girls. WC, BMI, pBMI%, and WHtR were similar between boys and girls (*p* > 0.05) except for HC, NC, and MUAC which were higher in girls than boys (*p* < 0.05). All anthropometric measures were higher in hypertensive participants compared to non-hypertensive participants (*p* < 0.05). Blood pressure measures were higher in hypertensive participants (*p* < 0.01), as well as SBP and HR which were higher in females than males (*p* < 0.05) (Table 1).

### 3.2. Relationship between Blood Pressure Measures and Obesity

Comparing blood pressure measures when participants were classified as obese using the standard pBMI cut-off at p95th (observed obesity) showed that the systolic blood pressure SBP and diastolic blood pressure DBP were higher (*p* < 0.05) in obese adolescents than non-obese adolescents. When individuals were classified as obese using pBMI cut-off at p85.2nd (predicted obesity), obese individuals had higher (*p* < 0.001) SBP and DBP than their non-obese counterparts. Observed obesity, defined by WHtR (0.481), showed higher SBP, DBP, and HR (*p* < 0.05) compared to their controls. Equally, obese individuals defined by HC, NC, MUAC, and NC showed higher SBP and DBP (*p* < 0.05) than their non-obese counterparts (Table 2).

### 3.3. Relationship between Hypertension and Obesity

There was an association between obesity and hypertension (Table 3). However, the association was higher for obese individuals defined by the predicted obesity cut-off (χ^2^ = 20.727; *p* < 0.001) than those defined by the observed obesity cut-off (χ^2^ = 8.227; *p* = 0.004). Similarly, the odds of developing hypertension were greater using the predicted obesity cut-off (OR: 2.070) compared to the predicted obesity cut-off value (OR: 1.748).

### 3.4. Performance of Blood Pressure Measures to Screen Obesity

The ROC analysis of blood pressure parameters using obesity cut-offs is summarised in Table 4. The ROC curve for SBP, DBP, HR, SBP%, and DBP% using pBMI85.2 as reference, illustrated in Figure 1A, showed a low AUC for BP measures ranging between 0.504 and 0.639 with SBP having the highest AUC. The sensitivity and specificity were highest for DBP% (0.657) and SBP% (0.624), respectively. The cut-off values to predict hypertension by SBP, DBP, HR, SBP%, and DBP% were 111.17; 71.17; 77.17; 65.5, and 70.25, respectively.

As shown in Figure 1B, the ROC curve for SBP, DBP, HR, SBP%, and DBP% using pBMI95 as reference showed a low AUC for BP measures which ranged between 0.504 and 0.642. SBP had the highest AUC, sensitivity (0.670), and specificity (0.586). The cut-off values to predict hypertension by SBP, DBP, HR, SBP%, and DBP% were 112.17; 71.83; 77.17; 63.85, and 71.80, respectively.

Using WHtR of 0.481 as the reference showed a low AUC for BP measures with SBP having the highest AUC of 0.621. SBP% had the highest sensitivity (0.701) and SBP had the best specificity (0.529). The predicted cut-off values for hypertension were 110.5; 70.50; 77.50; 50.5, and 69.9 for SBP, DBP, HR, SBP%, and DBP%, respectively (Figure 1C).

The ROC curve for SBP, DBP, HR, SBP%, and DBP% using a WHtR of 0.5 as the reference, illustrated in Figure 1D, showed a low AUC for BP measures with SBP having the highest AUC of 0.632. SBP% had the highest sensitivity and specificity of 0.699 and 0.615, respectively. The predicted cut-off values for hypertension were 111.8; 70.17; 77.17; 65.5, and 78.5 for SBP, DBP, HR, SBP%, and DBP%, respectively (Figure 1D).

Generally, obesity measures, both the WHO reference cut-off for WHtR of 0.5 and the CDC recommended pBMI% cut-off of 95%, as well as the new cut-offs for pBMI% of 85.2% and WHtR 0.48, performed poorly in predicting hypertension as the ROC curve showed the AUC below 70% for SBP percentile (SBP%) and DBP percentile (DBP%). The new cut offs for pBMI% and WHtR did not improve the sensitivity to predict hypertension as compared to the recommended cut-off values. The cut-off values for SBP% and DBP% from the ROC analysis ranged between the 60th and 70th percentile and were lower that the recommended value of 95 percentile.

## 4. Discussion

Various anthropometric indices used to screen obesity have been shown to predict hypertension in adulthood, among which a few have been suggested as an alternative for screening hypertension [16]. However, no consensus on anthropometric measures in screening childhood hypertension has been established. More so, with the increasing prevalence of hypertension in children and adolescents, there is a need for an alternative rapid diagnostic tool for hypertension in this population. This present study investigated the performance of obesity cut-offs established in our previous study to predict hypertension in adolescents in a South African population. The findings showed that the new obesity cut-off for pBMI and WHtR, as well as the CDC and WHO reference obesity cut-offs, had a poor performance in screening hypertension in this adolescent population, with a low AUC as well as low sensitivity and specificity. Hence, these obesity cut-offs had a poor discriminatory power for hypertension screening in adolescents.

Obesity has been considered a major risk factor for hypertension development both in adults [29] and children [30] and, therefore, may predict the presence of hypertension. However, the CDC and WHO recommended reference cut-off values of pBMI and WHtR, respectively, for obesity screening have been reported to differ across various race and ethnic population [31]. As such, this variation may affect the ability of these obesity screening tools to predict the diagnosis of hypertension across various ethnic populations. Based on the established obesity cut-off values for pBMI and WHtR in our previous report [21], which were lower than the recommend references by CDC and WHO, we hypothesized that they may predict hypertension differently. As such, we investigated the relationship of blood pressure and obesity defined by anthropometric cut-off values. Our findings showed that SBP and DBP were higher (*p* < 0.05) in obese adolescents compared to their non-obese counterparts. This is in accordance with previous studies which have shown obesity to be associated with hypertension in children [32,33] as well as in adolescents [34,35]. More so, obesity was strongly associated with hypertension (*p* < 0.01) with our predicted obesity (defined by pBMI85.2), showing a greater likelihood for hypertension (OR: 2.07; *p* < 0.01) than observed obesity defined by the recommended CDC reference of pBMI95 (OR: 1.748; *p* < 0.01). This finding suggests that the new pBMI% cut-off of 85.2% could be a better predictor of hypertension than the recommended CDC reference of 95%. The question which remains to be answered is if this new cut-off of obesity could serve as a bioindicator to predict hypertension screening in adolescents? This hypothesis was further investigated in this study using ROC analysis.

ROC analysis is commonly used to assess the performance of diagnostic screening tools as it provides the AUC as well as the sensitivity and specificity. In this case, the AUC evaluates the performance of anthropometric indicators for hypertension screening in adolescents. An AUC below 0.60 indicates a poor accuracy of obesity (pBMI and WHtR) cut-off in screening hypertension, whose occurrence may be random [36]. Moreover, an AUC above 0.9 is considered a good tool for its capacity to screen for hypertension. In this study, the AUC of pBMI and WHtR for screening DBP percentile and HR were <0.6 while the SBP percentile was 0.6 but ranged between 0.6 and 0.65. Similarly, the sensitivity of these cut-offs in predicting hypertension were between 50 and 60% for the HR and DBP percentile and between 60 and 70% for the SBP percentile. Equally, the sensitivity was low (<0.6) for all BP measures (DBP, SBP, and HR). There was no difference in the AUC, sensitivity, and specificity between the new cut-offs and the WHO and CDC reference cut-offs for pBMI and WHtR. Further, the cut-off values for blood pressure measures, as established from ROC analysis using anthropometric measures, were far below the recommended reference cut-off values for hypertension screening. This finding further confirmed obesity measures as a poor diagnostic tool for hypertension screening. This finding corroborates with a study involving Iranian children and adults which showed anthropometric indices with a poor accuracy for predicting elevated blood pressure and hypertension [37]. However, this contrasts with another study in Brazil which showed anthropometric indicators to have a good accuracy for high blood pressure screening in male adolescents [17]; meanwhile, studies in Spain [38] and Malaysia [16] showed anthropometric indices to have a moderate predictive capacity for high blood pressure in children and adolescents. Therefore, obesity measures may be considered a risk predictor rather than a screening tool for hypertension. This may be because obesity is an independent factor and not the sole risk factor for hypertension, as other factors such as high salts intake, alcohol, smoking, heredity, stress renal dysfunction, etc., present a risk for the development of hypertension [39,40]. This becomes even more complex as other factors, such as age, sex, and body composition, are related to blood pressure in children [41,42].

The strengths of this study, which supports these findings, are that a large population size was used and therefore backed by a strong predictive and statistical power. However, this study may be limited in that direct quantification of body fat was not used as a measure for obesity in predicting hypertension, considering that obesity is strongly associated with hypertension in the adult population. More so, other predictors of hypertension such as dyslipidaemia, smoking, hereditary, salt intake, etc., were not assessed. The study recruited predominately adolescents of Xhosa ethnicity and therefore generalization of the results to other populations in South Africa may not be applicable. The fact that the study was a cross-sectional study and other risk factors such as dietary intake, physical activity, and sedentary lifestyle, whose impact on hypertension depends on long-term monitoring, were not assessed limits the broader application of these findings.

## 5. Conclusions

Obesity established cut-offs for BMI percentile and WHtR in this populations showed poor performance in diagnosing hypertension even though they were strong predictors of hypertension. Therefore, obesity measures may not be sufficient to screen for hypertension in this adolescent population. However, obesity remains a significant risk to hypertension development and mandates regular monitoring in adolescents.

## Figures and Tables

**Figure 1 children-10-01520-f001:**
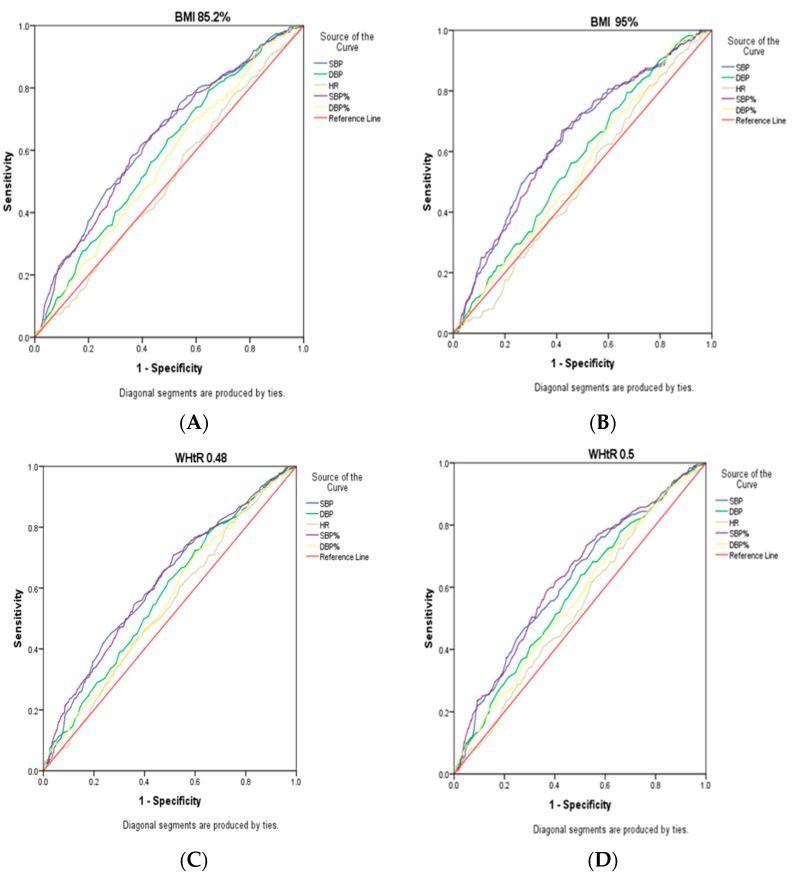
Receiver operating characteristic (ROC) curves for blood pressure measures using obesity measures (BMI% and WHtR) as reference (**A**–**D**).

**Table 1 children-10-01520-t001:** Baseline characteristics between hypertensive and non-hypertensive.

	Female		Male		*p*-Value		
	HT (136)	NT (660)	HT (57)	NT (291)	HT	Sex	HT × Sex
**Weight (kg)**	58.42 ± 14.13	54.75 ± 15.23	58.84 ± 16.29	49.43 ± 14.66	<0.001	0.283	0.211
**Height (m)**	1.56 ± 0.07	1.57 ± 0.08	1.61 ± 0.11	1.55 ± 0.11	0.019	0.723	0.043
**WC (cm)**	74.99 ± 10.19	72.52 ± 11.24	74.93 ± 11.34	69.79 ± 10.47	<0.001	0.825	0.341
**HC (cm)**	95.88 ± 12.42	91.26 ± 13.25	91.21 ± 13.69	86.23 ± 11.33	<0.001	0.036	0.547
**NC (cm)**	30.21 ± 1.98	29.84 ± 2.38	30.71 ± 3.45	29.87 ± 2.41	0.021	0.345	0.327
**MUAC (cm)**	25.29 ± 5.01	24.39 ± 4.31	24.51 ± 6.29	22.96 ± 4.65	0.015	0.027	0.519
**WHtR (cm)**	0.48 ± 0.06	0.46 ± 0.07	0.47 ± 0.07	0.45 ± 0.06	<0.001	0.818	0.910
**BMI (cm)**	23.73 ± 4.98	22.16 ± 5.33	22.35 ± 4.98	20.22 ± 4.61	<0.001	0.170	0.893
**pBMI (%)**	73.24 ± 26.71	63.10 ± 29.01	66.18 ± 29.80	54.95 ± 31.15	<0.001	0.003	0.827
**SBP (mmHg)**	125.47 ± 10.12	109.18 ± 9.84	127.16 ± 12.69	106.93 ± 11.56	<0.001	0.048	0.600
**DBP (mmHg)**	80.75 ± 7.18	69.71 ± 5.96	82.06 ± 6.76	69.84 ± 6.25	<0.001	0.565	0.870
**HR (** **bpm** **)**	84.31 ± 14.97	80.39 ± 11.39	82.95 ± 13.27	79.16 ± 12.01	<0.001	0.017	0.438

**Legend:** Mean ± SD; SD: standard deviation; NT: normotensive; HT: hypertensive; WC: waist circumference; HC: hip circumference; MUAC: mid-upper arm circumference; WHtR: waist to height ratio; BMI: body mass index; pBMI: BMI percentile; NC: neck circumference; SBP: systolic blood pressure; DBP: diastolic blood pressure; HR: heart rate.

**Table 2 children-10-01520-t002:** Comparison of blood pressure measures in children with or without obesity.

**Observed Obesity by pBMI at 95th Percentile**	**Non-Obese (967)**	**Observed Obesity (177)**	**t _(1142)_**	***p*-Value**
SBP (mmHg)	110.55 ± 12.36	116,29 ± 11.44	−5.749	<0.001
DBP (mmHg)	71.37 ± 7.61	73.16 ± 6.91	−2.912	0.004
HR (bpm)	80.67 ± 12.45	80.72 ± 10.78	−0.052	0.958
**Predicted Obesity by pBMI at 85.2nd Percentile**	**Non-Obese (781)**	**Predicted Obesity (352)**	**t _(1133)_**	** *p* ** **-Value**
SBP (mmHg)	109.7021	115.43 ± 11.99	7.350	<0.001
DBP (mmHg)	70.96 ± 7.63	73.18 ± 7.12	4.599	<0.001
HR (bpm)	80.67 ± 12.37	80.77 ± 11.87	0.131	0.896
**Predicted Obesity by WHtR at 0.481**	**Non-Obese (782)**	**Predicted Obesity (362)**	**t _(1144)_**	** *p* ** **-Value**
SBP (mmHg)	109.79 ± 12.08	114.99 ± 12.34	6.727	<0.001
DBP (mmHg)	70.91 ± 7.40	73.26 ± 7.57	4.912	<0.001
HR (bpm)	80.13 ± 12.35	81.86 ± 11.81	2.235	0.026
**Predicted Obesity by WHtR at 0.5**	**Non-Obese (855)**	**Predicted Obesity (289)**	**t _(1142)_**	** *p* ** **-Value**
SBP (mmHg)	110.1856 ± 12.27	115.16 ± 12.03	−5.986	<0.001
DBP (mmHg)	71.06 ± 7.46	73.38 ± 7.48	−4.551	<0.001
HR (bpm)	80.42 ± 12.39	81.43 ± 11.61	−1.210	0.226
**Predicted Obesity by WC at 75.1 cm**	**Non-Obese (787)**	**Predicted Obesity (357)**	**t _(1134)_**	** *p* ** **-Value**
SBP (mmHg)	109.16 ± 12.03	116.48 ± 11.68	9.619	<0.001
DBP (mmHg)	70.77 ± 7.52	73.58 ± 7.21	5.916	<0.001
HR (bpm)	80.73 ± 12.44	80.55 ± 11.69	−0.230	0.818
**Predicted Obesity by HC at 92.2 cm**	**Non-Obese (705)**	**Predicted Obesity (438)**	**t _(1133)_**	** *p* ** **-Value**
SBP (mmHg)	108.32 ± 11.72	116.49 ± 11.79	11.434	<0.001
DBP (mmHg)	70.69 ± 7.58	73.20 ± 7.19	5.532	<0.001
HR (bpm)	80.82 ± 12.48	80.46 ± 11.77	−0.491	0.628
**Predicted Obesity by MUAC at 26 cm**	**Non-Obese (513)**	**Predicted Obesity (249)**	**t _(762)_**	** *p* ** **-Value**
SBP (mmHg)	105.57 ± 10.61	112.08 ± 12.35	7.521	<0.001
DBP (mmHg)	71.07 ± 7.56	72.77 ± 7.56	2.907	0.004
HR (bpm)	81.57 ± 12.62	82.88 ± 12.34	1.359	0.178
**Predicted Obesity by NC at 30.6 cm**	**Non-Obese (501)**	**Predicted Obesity (263)**	**t _(764)_**	** *p* ** **-Value**
SBP (mmHg)	105.31 ± 11.17	112.32 ± 11.05	8.269	<0.001
DBP (mmHg)	70.99 ± 7.68	72.85 ± 7.27	3.217	0.001
HR (bpm)	82.23 ± 12.56	81.57 ± 12.45	−0.688	0.492

**Legend:** Mean ± SD; SD: standard deviation; SBP: systolic blood pressure; DBP: diastolic blood pressure; HR: heart rate. WC: waist circumference; HC: hip circumference; MUAC: mid-upper arm circumference; WHtR: waist to height ratio; pBMI: body mass index percentile; NC: neck circumference.

**Table 3 children-10-01520-t003:** Relationship between obesity and hypertension.

		Normotensive	Hypertensive	Total	Chi-Square	OR	*p*-Value
Observed	Non-obese	817 (85.9)	150 (77.7)	967	8.227	1.748	0.004
Obesity	Obese	134 (14.1)	43 (22.3)	177			
(pBMI at p95th)	Total	951	193	1144			
Predicted	Non-obese	685 (72.0)	107 (55.4)	792	20.727	2.070	<0.001
Obesity	Obese	266 (28.0)	86 (44.6)	352			
(pBMI at p85th)	Total	951	193	1144			

**Table 4 children-10-01520-t004:** Optimal cut-off, sensitivity, specificity, SE, and area under the ROC curves for blood pressure measures in predicting hypertension using pBMI and WHtR.

	Cut-Off	Sensitivity	1-Specificity	AUC	SE	95%CI
**Ref: pBMI 85.2**						
SBP	111.17	0.651	0.439	0.639	0.018	0.605–0.674
DBP	71.17	0.577	0.455	0.588	0.018	0.553–0.623
HR	77.17	0.597	0.465	0.504	0.018	0.468–0.540
SBP%	65.5	0.589	0.376	0.635	0.018	0.601–0.670
DBP%	70.25	0.657	0.553	0.560	0.018	0.524–0.595
**Ref: pBMI 95**						
SBP	112.17	0.670	0.434	0.642	0.022	0.599–0.686
DBP	71.83	0.563	0.456	0.569	0.022	0.526–0.612
HR	77.17	0.608	0.569	0.504	0.022	0.460–0.547
SBP%	63.85	0.670	0.414	0.641	0.022	0.598–0.685
DBP%	71.80	0.614	0.551	0.543	0.022	0.499–0.587
**Ref: WHtR 0.48**						
SBP	110.5	0.650	0.471	0.615	0.018	0.583–0.657
DBP	70.50	0.624	0.495	0.579	0.018	0.545–0.619
HR	77.50	0.611	0.549	0.538	0.018	0.500–0.574
SBP%	50.50	0.708	0.515	0.621	0.017	0.596–0.669
DBP%	69.90	0.662	0.559	0.557	0.018	0.527–0.602
**Ref: WHtR 0.5**						
SBP	111.83	0.613	0.443	0.620	0.019	0.583–0.657
DBP	70.17	0.642	0.517	0.582	0.019	0.545–0.619
HR	77.17	0.583	0.533	0.537	0.019	0.500–0.574
SBP%	65.50	0.699	0.385	0.632	0.019	0.596–0.669
DBP%	78.50	0.500	0.404	0.564	0.019	0.527–0.602

**Legend:** WHtR: waist to height ratio; pBMI: body mass index percentile; SBP: systolic blood pressure; DBP: diastolic blood pressure; HR: heart rate; AUC: area under the curve; SE: standard error; CI: confidence interval.

## Data Availability

All participant’s data are kept confidential as per the South Africa National Data Protection Act. However, data may be made available upon reasonable request to the author of correspondence.

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
