# Peer review of "Assessment of the Performance of Obesity Measures in the Screening of Hypertension in a South African Adolescent Population"

_children, 2023, doi:10.3390/children10091520_

Round 1

Reviewer 1 Report

This study tested the ability of various obesity cut-offs to screen for hypertension among  n=1144 11-17 year olds from the Eastern Cape Province in South Africa.  While obesity was strongly related to hypertension, obesity cut-offs (defined in various ways using BMI, waist circumference and waist-to-height ratio) did only a modest job of predicting hypertension, with all areas under the receiver-operating characteristic curves <0.65.

This is a well written paper that has been appropriately designed and analysed.  Minor typos aside, I have no comments for the authors and commend them on an excellent paper.

Typos.

p1. First line of introduction. "continuous" should be "continues".

p3. line 118. "analysed" should be 'analyse".

Author Response

Dear Reviewer,

I have addressed your comments highlighted in red colour in the manuscript.

p1. First line of introduction. "continuous" has been changed to "continues".

p3. line 118. "analysed" has been changed to 'analyse".

Reviewer 2 Report

This discussion and conclusion should introduce the need for a novel test developed by the authors to better predict the risk for hypertension in adolescent populations.
The follow-up publication should begin by referencing a few of the more than three thousand articles linking Maillard reaction products (fast-food, convenience food, processed food, baked, grilled, fried, broiled, barbequed, canned, bagged, boxed, bottled, grain-fed, conventional, etc.) and hypertension; Maillard reaction products (MRP) and oxidative stress; and oxidative stress and hypertension.
Design four groups approximating food and beverage calorie consumption daily. For example, Hypertension Risk O = Processed Food (PF) consumption less than 25% of total daily calories consumed; 1 = 26 to 50%; 2 = 51 to 75%; and 3 = 76 to 100%.
The expected reliability of the new MRP-HTN Risk measure for adults and adolescents, or whatever the authors call the new tool, can be discussed as superior to traditional BMI and related standards, especially in young adults and adolescents.
Younger people have more enormous reserves of stem cells known to be rendered useless by MRP-induced inflamed, space-occupying, differentiated, and undifferentiated stem cells misperceived as overweight from excess non-MRP calories. Dietary MRP temperatures and conditions denature the wholesome nutrients stem and differentiated cells need to function into MRP-antigen. Extra calories from non-MRP food and beverages do not readily lead to overweight or hypertension. MRP food and drink contain higher phosphorus from degraded membrane phospholipids and mitochondrial ATP. Phosphorus-rich food browning instantly catabolizes immune-strengthening glutamine to immune-weakening glutamate. Glutamate is the same central nervous system stimulant surged by cocaine, methamphetamine, and Adderall. Hence, highly habit-forming and alluring dietary MRPs directly trigger hyperphosphatemia, hyperammonemia, systemic redox imbalance, immune dysregulation, systemic inflammation of trillions of cells misunderstood as overweight and metabolic syndrome, directly producing hypertension.
The higher relative body percentage of stem cells in younger people is less likely to predict hypertension accurately based on BMI and related measures. Because the propensity for increased systemic cellular inflammation, misdiagnosed as obesity, more often takes decades to evolve.

The suggested literary edits for the Conclusion appear below and illustrate that the article may need additional literary improvement:

Obesity-established cut-offs for BMI percentile and WHtR in these populations showed poor performance in diagnosing hypertension even though they were strong predictors of hypertension. Therefore, obesity measures may not be sufficient to screen for hypertension in this adolescent population. However, obesity remains a significant risk to hypertension development and mandates regular adolescent monitoring.

This and the follow-up article have the potential of helping millions of people live longer, more accomplished lives.

See above

Author Response

Dear reviewer,

The authors have addresed your first concern by updating the the first paragraph of the discussion with a background of the study which introduces the need for a new diagnostic test.

However, we think there is a mix-up in your subsequent suggestion which talks about Maillard reaction products (MRP). This research has nothing to do with MRP so we are not making any changes in the manuscript with regards to MRP.

the authors have equally done the corrections suggested in the conclusion.

All corrections are highlighed in red colour

Thank you